# Promoting participation in physical activity through Snacktivity: A qualitative mixed methods study

Matthew Krouwel[1], Sheila M. Greenfield[2], Anna Chalkley[3], James P. Sanders[1], Helen M. Parretti[4], Kajal Gokal[1], Kate Jolly[2], Magdalena Skrybant[2], Stuart J. H. Biddle[5], Colin Greaves[6], Ralph Maddison[7], Nanette Mutrie[8], Natalie Ives[9], Dale W. Esliger[10], Lauren Sherar[10], Charlotte L. Edwardson[11], Tom Yates[11], Emma Frew[12], Sarah Tearne[9], Amanda J. Daley[1]*

1 Centre for Lifestyle Medicine and Behaviour, School of Sport, Exercise and Health Sciences, Loughborough University, Loughborough, United Kingdom, 2 Institute for Applied Health Research, University of Birmingham, Birmingham, United Kingdom, 3 Centre for Applied Education Research, Wolfson Centre for Applied Health Research, Bradford Royal Infirmary, Bradford, West Yorkshire, United Kingdom, 4 Norwich Medical School, Faculty of Medicine and Health, University of East Anglia, Norwich, United Kingdom, 5 University of Southern Queensland, Springfield, Australia and Faculty of Sport & Health Sciences, University of Jyväskylä, Jyväskylä, Finland, 6 School of Sport, Exercise and Rehabilitation Sciences, University of Birmingham, Birmingham, United Kingdom, 7 Institute for Physical Activity and Nutrition, Deakin University, Melbourne, Australia, 8 Physical Activity for Health Research Centre, University of Edinburgh, Edinburgh, United Kingdom, 9 Birmingham Clinical Trials Unit, Institute for Applied Health Research, University of Birmingham, Birmingham, United Kingdom, 10 School of Sport, Exercise and Health Sciences, Loughborough University, Loughborough, United Kingdom, 11 Diabetes Research Centre, College of Life Sciences, University of Leicester and NIHR Leicester Biomedical Research Centre, Leicester, United Kingdom, 12 Health Economics Unit, Institute for Applied Health Research, University of Birmingham, Birmingham, United Kingdom

* a.daley@lboro.ac.uk

**Data Availability Statement:** All relevant data are within the paper and its Supporting information files, as well as available from figshare at https://figshare.com/s/0eea6f01a8f340abbe64.

## Abstract

### Background

Public health guidance acknowledges the benefits of physical activity of any duration. We have proposed a whole-day approach to promoting physical activity called Snacktivity™, which encourages frequent 2–5 minute 'activity snacks' of moderate-to-vigorous intensity.

### Methods

Using repeated semi-structured interviews and a think aloud protocol, this study aimed to understand participants' experiences of integrating Snacktivity™ into daily life, to provide insights to refine the delivery of Snacktivity™ interventions. Physically inactive adults recruited via primary care and a community health service engaged with an intervention to encourage Snacktivity™ over three weeks, which included using a Fitbit and linked mobile phone app (SnackApp). Participants took part in semi-structured interviews on two occasions during the intervention, with a sub-group participating in a think aloud study. Three study data sets were generated and independently explored using inductive thematic analysis, with findings combined into a single set of themes.

**Funding:** This work was financially supported by the National Institute for Health Research (NIHR) (www.nihr.ac.uk.) in the form of a grant (RP-PG-0618-20008). AJD was supported by National Institute for Health Research (NIHR) (www.nihr.ac.uk.) in the form of a Research Professorship award. The views expressed are those of the authors and not necessarily those of the NHS, the NIHR, or the Department of Health and Social Care. The funders had no role in study design, data collection and analysis, decision to publish, or preparation of the manuscript.

**Competing interests:** The authors have declared that they have no competing interests.

## Results

Eleven adults participated in the interview study who were interviewed twice (total interviews completed n = 21, 1 participant declined the second interview), of whom six completed the think aloud study (total voice recordings n = 103). Three main themes emerged from the combined data; lived experience of participating in Snacktivity™, motivation for Snacktivity™ and experiences with the Snacktivity™ technology. Participants undertook a variety of activity snacks, utilising their environment, which they believed improved their psychological wellbeing. Participants were enthusiastic about Snacktivity™, with some stating that activity snacks were more accessible than traditional exercise, but perceived they were often prevented from doing so in the presence of others. Participants were mostly enthusiastic about using the Snacktivity™ technology.

## Conclusion

Participants were able to incorporate Snacktivity™ into their lives, particularly at home, and found this approach acceptable. Participants felt they experienced health benefits from Snacktivity™ although barriers to participation were reported. This study offers insights for translating guidance into practice and supporting people to become more physically active.

## Introduction

Many people find it difficult to engage in regular physical activity and over 40% of the population are insufficiently active to optimally benefit their health [1]. If the World Health Organisation's Global Action Plan on Physical Activity goal of a 15% reduction in physical inactivity is to be met by 2030, concerted actions and policies are needed now [2]. Guidance for physical activity states that over a week, at least 150 of moderate intensity or 75 minutes of vigorous intensity physical activity (MVPA) should be accumulated, which has often been promoted as a minimum of 30 minutes on at least 5 days per week [3]. However, revised guidelines now acknowledge the health benefits of regular intermittent bouts of MVPA of any duration [3]. To support the updated guidelines, we have proposed an alternative 'whole day' approach to promoting physical activity called Snacktivity™, which encourages people to take short but frequent 'activity snacks' of 2–5 minutes of MVPA throughout the day, to achieve at least 150 minutes of MVPA per week [4]. Experimental studies support the position that continuous and intermittent MVPA for the same total duration leads to similar health outcomes, although this is yet to be examined in real world contexts [5, 6].

A detailed rationale for Snacktivity™ has been published previously [4]. Briefly, Snacktivity™ can be completed throughout the day whilst undertaking daily tasks, (i.e., taking the stairs instead of the lift, squats whilst waiting for the kettle to boil) and therefore has the potential to address a common barrier to physical activity, often reported a perceived lack of time [7]. Small changes are easier for people to initiate and maintain than large changes [8], therefore the promotion of short bouts of physical activity through Snacktivity™ may help to develop people's confidence to become physically active more regularly. Adults typically spend a substantial part (i.e. 60–70%) of their waking day sedentary which may have adverse effects on their health [9]. An important benefit of Snacktivity™ over current physical activity guidance is that it could address concerns about sedentary time by encouraging the break-up of passive sitting time throughout the whole day. Physical activity guidance also recommends that the public participate in muscle strengthening activities at least twice per week and many of these

types of activities can be achieved through short, frequent bouts of physical activity (e.g. seated leg raises and squats) [3]. Snacktivity™ places equal emphasis on aerobic and strength-based activity.

Whilst observational studies have shown that Snacktivity™ is an acceptable approach to promoting physical activity [10, 11], it is not yet clear how, or if people can maintain Snacktivity™ over time. The strategies people may use to facilitate their daily participation in Snacktivity™ and their success in achieving participation in regular Snacktivity™ is also yet to be determined. The day-to-day challenges that people may face in completing Snacktivity™ are also unclear. This study aimed to understand and capture peoples' experiences of participating in Snacktivity™ using repeated semi-structured interviews to understand changes over time, and explore participants' 'in the moment' engagement with Snacktivity™, using a think aloud protocol. A key aim was to gather qualitative data to facilitate the further development of the Snacktivity™ concept, including the useability of technology developed to support participation in Snacktivity™.

## Materials and methods

### Study context

This study was conducted between January and June 2021, which included some of the COVID-19 lockdown periods in the UK therefore the methods were adapted to comply with UK law at the time. Favourable ethical opinion was obtained from NRES Research Ethics Committee London–Surrey Research Ethics Committee (REF:20/PR/0589). Participants received a £20 shopping voucher for taking part and for any out-of-pocket expenses. The research was undertaken by a diverse team, including a specialist in behavioural medicine (AD), a medical sociologist (SG), a health scientist (MK) and a physical activity academic (AC). This research was supported and guided by a Patient and Public Advisory Group. The study is reported in line with the consolidated criteria for reporting qualitative research checklist (S1 Checklist).

### Recruitment of participants

A convenience sample of adults receiving treatment from a National Health Service (NHS) podiatry service in the West Midlands (UK) were recruited by an invitation letter from the service. Adults who had completed a Snacktivity™ study survey [11] via an invitation from their general practice as part of a prior Snacktivity™ study, and who had consented to be contacted to participate in future Snacktivity™ studies, were also invited by email to take part. Those interested in participating contacted the research team directly and were given a participant information sheet and asked to provide written informed consent to participate.

### Screening, eligibility and consent

Responders who were inactive or moderately inactive using the General Practice Physical Activity Questionnaire were eligible to participate [12]. Participants needed to own a smartphone with either iOS 10+ or Android OS 4.0+, able to provide written informed consent and understand the English language sufficiently to complete the study procedures. Participants were excluded if they were unable to provide informed written consent or did not consent that their healthcare provider was informed of their involvement in the study. Participants were able to consent to either the interview study, the think aloud study or both. As the participant pool for this study was modest, it was decided that all participants who consented would be

interviewed and data would be analysed for maximum richness. As part of the recruitment procedures participants gave the research team their personal contact details.

## Snacktivity™ intervention

The Snacktivity™ intervention is based on self-regulation theory and the habit formation model, which focuses on promoting self-monitoring and self-regulatory techniques (e.g. goal setting and behavioural feedback) to facilitate habit formation [13–16]. Self-monitoring techniques encourage tracking, which is important to develop Snacktivity™ habits and also provides participants with the ability to record and track the number of activity snacks they might typically perform each day. The behavioural goal was for participants to work towards incorporating ≥150 minutes of Snacktivity™ into their daily lives for three weeks.

The NHS initiative 'Making Every Contact Count' in England aims to promote routine conversations about health behaviour change between health care professionals and their patients [17]. In accordance with Making Every Contact Count, the Snacktivity™ intervention was primarily designed as a brief intervention (~5–7 minutes) to be prompted/raised by health care professionals within routine consultations. Those recruited via the podiatry service had their Snacktivity™ consultation with their treating podiatrist and those recruited from the prior Snacktivity™ survey study received a standalone consultation with a general practitioner (GP) who is part of the research team. During the consultations, the podiatrist/GP raised the topic of the Snacktivity™ study and explained the purpose and benefits of Snacktivity™. The health care professionals encouraged engagement with the supporting technology, a mobile phone app (SnackApp), which linked to a wrist-worn Fitbit Versa 2 activity monitor provided free of charge to participants. The Fitbit displayed a bespoke Snacktivity™ clockface, which provided participants with the number of completed activity snacks, steps, and active minutes (MVPA) they had completed each day. This technology allowed participants to monitor their Snacktivity™/physical activity, receive feedback on progress and push notifications to remind, motivate and encourage Snacktivity™, as well as the opportunity to form action plans for Snacktivity™. A printed copy of the Snacktivity™ picture board that illustrates a variety of activity snacks was also provided to participants by health care professionals. All health care professionals attended a one-hour intervention training session delivered by AD. The Snacktivity™ consultations were delivered following a standard consultation checklist to ensure intervention fidelity.

## Semi-structured interviews

Two sets of semi-structured interviews (interview 1 and interview 2) were conducted by a female post-doctoral qualitative researcher (AC) with five years of experience who was already known to the participants as a researcher with an interest in health behaviours. For this study AC favoured a pragmatic philosophical stance, intended to recognize the gap between academic and real-world applied practice and the need to reduce this gap by being problem centred. The interview schedule was not piloted as it was based on prior work [18] (see S1 File). The interviews were conducted at the end of weeks 1 and 2 of the Snacktivity™ intervention, with the same participants for both interviews, and aimed to capture and explore changes in participants' thoughts, behaviours, and views over the study time period. The semi-structured interviews, which consisted of a series of pre-determined topic-orientated and predominantly open-ended questions, provided the opportunity for participants to talk more widely, whilst remaining relevant to the subject of interest [19]. A policy of allowing the first interview to be succinct was followed to avoid overburdening participants and thus reducing the likelihood of them declining to participate in the second interview. Interviews were conducted by telephone

to comply with COVID-19 restrictions at the time and audio recorded using encrypted voice recorders. Brief field notes were made after each interview capturing any relevant contextual information and reflections on discussions, however formal records were not kept of participants' location or the presence of other people during the interview.

## Think aloud study

The think-aloud study took place each day across the three-week Snacktivity™ intervention period. A think aloud methodology provides specific opportunity to capture participants' ongoing thought processes during an experience, allowing researchers to analyse users' reactions to an intervention and the cognitive processes and behaviours being performed, as they occur [20, 21]. Participants were asked to record their impressions, thoughts, and feelings about Snacktivity™ on their mobile phone as 'voice notes' during or shortly after completing their activity snacks, and these were then sent to the research team whenever participants chose to do so each day using the WhatsApp digital platform. Participants were asked not to filter their thoughts, but to share everything they had to say about their Snacktivity™ experience. Participants were reminded every day (including weekends) by text message to send their voice notes each day.

## Data analysis

A multi-person team approach to data analysis was used for enhanced trustworthiness [22]. The same approach to analysis was used for all three data sets (interviews 1–2 & think aloud voice notes), but they were analysed separately, eventually combined, and reported as one. This approach to data analysis was taken because with several interconnected data sets with multiple data points at different times, performing the analyses separate would be more likely to highlight difference between data sets than if they were analysed at as a single data set. The interviews and think-aloud voice recordings were transcribed independently and not shared outside of the team, equally the coding was not discussed beyond the coding team (AD, MK & SG). Transcriptions were analysed by inductive framework analysis to identify naturally occurring themes [23]. Data management was facilitated by using the NVivo12 software package. The first three participant recordings were coded (MK) and initial themes were identified, critically discussed, and agreed upon within the team (MK & SG), adjustments were made, and the remaining data were coded [24]. Any proposed changes to the themes thereafter were discussed within the wider team (AD, MK & SG).

## Results

### Recruitment and data

A total of 20 eligible individuals agreed to take part (6 men & 14 women) and 11 participants took part in interview 1 (week 1), 10 in interview 2 (week 2) (total interviews = 21 as one participant declined interview 2) (Table 1), and six in the think-aloud study (1 man & 5 women) (Table 2). On average, interviews conducted in week 1 were 16.5 minutes in duration (range 11–33 minutes) and interviews conducted in week 2 were 15.8 minutes (range 10–27 minutes). The think aloud study data produced an average of 17 recordings per participant over the intervention period (range 14–21 recordings). Seven participants had their initial Snacktivity™ intervention consultation with their podiatrist and three with the study GP. Due to operational difficulties during COVID-19, one consultation due to take place with a podiatrist was delivered by a member of the research team.

Table 1.  Interview participant demographics.

| Participant demographics | Week 1 | Week 2 |
|---|---|---|
| | N = 11 | N = 10 |
| Mean age (years/SD) | | |
| | 62.3 (9.5) | 60.0 (6.07) |
| Gender | | |
| Male | 3 (27.3%) | 2 (20.0%) |
| Female | 8 (72.7%) | 8 (80.0%) |
| Employment status | | |
| In full time paid employment | 2 (18.2%) | 2 (20.0%) |
| In part time paid employment | 4 (36.4%) | 4 (40.0%) |
| Retired from paid work | 5 (45.5%) | 4 (40.0%) |
| Ethnicity | | |
| White | 10 (90.9%) | 9 (90.0%) |
| Other (Indian) | 1 (9.1%) | 1 (10.0%) |
| Mean BMI (kg/m$^2$ /SD) | | |
| | 26.5 (3.83) | 26.9 (3.7) |
| BMI Classification | | |
| Healthy (18.5–24.9) | 5 (45.5%) | 4 (40.0%) |
| Overweight (25–29.9) | 4 (36.4%) | 4 (40.0%) |
| Obese (30+) | 2 (18.2%) | 2 (40.0%) |
| GPPAQ Classification | | |
| Inactive | 9 (81.8%) | 8 (80.0%) |
| Moderately inactive | 1 (9.1%) | 1 (10.0%) |
| Moderately active | 1 (9.1%) | 1 (10.0%) |

## Themes

The three studies generated three sets of themes and subthemes relevant to the aims of this report. These themes and subthemes demonstrated substantial similarity across all three studies therefore the decision was made to combine them into a single set of themes at the theme and subtheme level. S2 File provides further detailed information regarding theme generation according to each data set and the process by which the combined data set presented in Table 3 was generated. One theme identified in the second set of interviews named 'awareness and impact of being a participant in a health study', was not included in the combined themes as it was considered to be primarily of interest to the research team in the design of future Snacktivity™ trials. The final themes consisted of the lived experience of participating in Snacktivity™, experiences with the Snacktivity™ technology and motivation for Snacktivity™ (Table 3).

**The lived experience of participating in Snacktivity™.**   This theme sees a developmental journey from the introduction of the Snacktivity™ concept, to the behaviours this led to and created, to the effect the behaviours had on participants lives.

## Embracing Snacktivity™

Participants engaged with the concept of Snacktivity™, appearing to understand the idea of short bouts of physical activity, and they acknowledged that this was a useful route to achieving meaningful participation in physical activity each day.

**Table 2. Think aloud protocol participants.**

| Participant demographics | All participants, N = 6 |
|---|---|
| Mean age (years/SD) | |
| | 59.1 (6.4) |
| Gender | |
| Male | 0 (0%) |
| Female | 7 (100%) |
| Employment status | |
| In full time paid employment | 2(28.6%) |
| In part time paid employment | 1 (14.3%) |
| Retired | 4 (57.1%) |
| Ethnicity | |
| White | 6 (85.7%) |
| Indian | 1 (14.3%) |
| Mean BMI (kg/m$^2$ /SD) | |
| | 27.7 (3.8) |
| BMI classification | |
| Healthy | 2 (28.6%) |
| Overweight | 2 (28.6%) |
| Obese | 3 (42.9%) |
| GPPAQ classification | |
| Inactive | 5 (71.4%) |
| Moderately inactive | 1 (14.3%) |
| Moderately active | 1 (14.3%) |

"You know people always think, you know, you always think, oh you know you need to go and do a twenty minute workout, and I think you know what you've shown us is that we can you know choose to do whatever, whatever it is that you enjoy doing, just . . . just have that moment for two minutes, you know, it doesn't even have to be five. . ."

(Int2-2006)

"That's what I am doing, I'm doing little snacktvities in between times,"

(TA-2021)

One subtle indication of how much some participants engaged with Snacktivity™ was their use of its terminology, specifically referring to 'snacks' as brief bouts of physical activity.

"Right, I've just done some stair climbing, pleased with that. Got quite puffed by the end of it and my heart rate went up so obviously I've been working very hard at that stair climbing. Yeah, I think today's gone all right I've done about 6,000 steps and I ate different [activity] snacks, including two I haven't recorded, because I forgot."

(TA-2004)

## Locations for Snacktivity™

For many participants the home, which appeared to be the main location for conducting activity snacks, and it was used in varied ways, in particular the steps/stairs, which participants walked or ran up and down, or used as a platform for step-ups or press-ups. The kitchen was

**Table 3. Themes and subthemes across all three data sets and combined themes.**

| Theme | Definition | Subtheme | Definition | Codes |
|---|---|---|---|---|
| **The lived experience of participating in Snacktivity™** | The experience, be that physical, social or psychological of conducting snacks, participating in Snacktivity™ | Embracing | The understanding of and engagement with Snacktivity™ | Embracing statements and statements of intent |
| | | Location | Where and when snacks are conducted. | Location, snack location, home gym |
| | | Type, variety and intensity | What snacks were done and what influenced the choice | Type, type of exercise, type of activity, reason for choice of Snack, limitations upon snacks |
| | | Experience and impact | The lived experience of and reflections upon doing Snacktivity™, and the noted impact upon physical, psychological, and emotional wellbeing | Effects of (general), Positive statements and noticeable benefits |
| **Motivation for Snacktivity™** | Reasons for conducting and persevering with activity in general and Snacktivity™ specifically | Barriers and demotivation | Limitations upon the participants ability to conduct snacks or to persist in engaging with Snacktivity™, be they physical, mental, familial, social, financial or anything else | Motivation—positive, Habit building, Recognition of starting point, Physical activity persistence, Physical activity intent, Negative comments, Self-assessment of ongoing activity and Snacktivity™ behaviour, Non-Snacktivity™ prompts to physical activity, Barriers and limitations, Motivational thoughts, Evidence of persistence, Limitations and promoting comments, Demotivational experience and comments, Positive motivational, Assessment of activity, Reappraisal of activity, Improvement (general), Snacktivity™ improvements, Low activity, SnackApp demotivation |
| | | Routine and habit formation | Evidence of the establishment of Snacktivity™ or physical activity habits | Habit and routine, establishing routine |
| **Experiences with Technology Snacktivity™** | Participants summations, commentary, assessments and reactions to the technology of Snacktivity™ | Fitbit and SnackApp | Comments about and thoughts upon the Fitbit activity tracker, the SnackApp, their various features and the interaction of the two | Fitbit, SnackApp, Fitbit—value, judgement, Snack App—value judgement, Suggested improvements, Lack of understanding of the SnackApp, Negative comments about the SnackApp, Device working properly, Fitbit general and positive, Fitbit negative, Fitbit improvements, Problems, Familiarisation, SnakApp use, Forum use, Suggested improvements |
| | | Notification, alerts and prompts | Reactions to and commentary on reminders to conduct snacks, congratulatory notifications and other automated communications | Fitbit—value judgement, SnackApp—value judgement, Snacktivity™ prompts activity, Reflections upon prompts, Notification motivation, More snacks and prompting |

also seen as a useful location with participants dancing in it, or using the tiled floor as a hopscotch court. Others used the edge of a counter or sink as a balance for squats and heel raises. Living room areas provided soft furnishings which could reduce the pressure on hands whilst doing weight bearing activities. Participants also improvised with strength-based activities, using household objects to add resistance to arm exercises.

"...straight after I'd done the push-ups on the stairs, I grabbed my tins of beans and done the ... a sort of set of what I used to do ... like I think lateral raises and forward raises and sort of shoulder presses and stuff with my tins of beans."

(Int2–2023)

"did some quick steps in the kitchen, used the tile squares to do quick steps from one square to the other square and did that activity."

(TA-2006)

"I'll go and do some star jumps in the lounge you know."

(TA-2005)

"So for me, and we've got a big house so for me, doing. . . I'm doing quite a lot of housework now that I didn't do before"

(Int1-2005)

## Type, variety and intensity of activity snacks

Across all recordings participants described taking part in thirty different activity snacks which took many forms (Table 4). The most popular activity snacks were walking, housework, on-the-spot activities like marching or walking and ascending/descending stairs carried out in such a way that increased the activity/effort/intensity. Some activity snacks were achieved by adapting everyday tasks, such as parking further away from the destination than normal, or deliberately swinging legs whilst cooking.

"I am about to go and hang out my washing and what I am going to do, I've got a fairly decent sized garden and I'm going to leave the basket at one side and I'm going to take one item at a time and as I get further and further away from the basket, I'm going to run back to the basket."

(TA-2023)

"I do my ironing, and instead of just putting a pile of you know . . . accumulating a pile of ironed T-shirts on the side, I'm walking in between each one and putting them down else-where, you know, that sort of thing."

(Int1-2004)

There was also a sense of a diminishing variety of activity snacks performed as the intervention progressed, however no participants expressed that they felt they were doing fewer activity snacks as a consequence.

"I do feel as if I keep doing the same thing because I can't think what else to do."

(TA-2014)

"I feel as though I've settled into it a bit more now. . . . . .initially I didn't know what to do, and I did all sorts of jumping about and running around"

(Int2-2014)

**Table 4. Activity snacks completed by participants.**

arm curls, arm flailing, arm raises, calf raises, dance and dance like activity, elliptical machine, gardening, guided workout, heel raises, hopscotch, housework and house work enhanced with additional effort/movement, jogging, jumps, knee/leg raises, lunges, mountain climbers, moving on the spot (marching, stepping or walking), Pilates, press ups (against stairs/chair/wall), running, sideways steps, sit ups and half sit ups, squats, and squats with support, stair climbing (ascending and descending), standing up, step ups, tensing/flexing muscles, stretches, treadmill, virtual reality sports/ activity, walking.

## Experience and impact

Several participants expressed noticeable benefits from completing their activity snacks, including experiencing a positive mental state and perceived physical benefits to their health.

"It might have been yesterday, when it crossed my mind, I feel well inside, and I actually do feel well. As I say, I'm going through the menopause, so there are days I don't feel great mentally, but actually for the first time in a long time I feel healthy. So, I do think it's having an impact, I really do think it's having an impact."

(Int2-2023)

"Now doing press ups against the stairs. It's quite hard work on the arms and increases breathing a little bit I think, bound to be doing me good."

(TA-2010)

**Motivation for Snacktivity™.** Many of the participants indicated that they had found the Snacktivity™ study motivational. Some of these expressions are related to the short bursts, 'snack', approach to physical activity.

"I'm not going to be one of these you know go and spend hours in the gym, but you know these sort of things are well within your means and I can do them . . .because you know sometimes you're not just motivated or . . . but it's not going to take much to do two minutes of something."

(Int2-2006)

## Barriers and demotivation

Some participants encountered perceived challenges to conducting Snacktivity™, such as a lack of space. Work appeared to be problematic for some, with the implication that social norms for the workplace do not allow for 'snacks'

"But yeah and I think you know perhaps when we're back at the office for people, you know you couldn't if you were you know in a meeting or even just working at your desk, you couldn't you know necessarily get up and say, oh well I've got to move, could you?"

(Int1-2012)

One participant experienced pressure from their partner to use the stairs for 'snacks' less.

". . . you know he says, oh God I can hear you on those stairs again!"

(Int2-2014)

And another felt that they could not conduct snacks whilst on video calls. which suggests that both an explicit and assumed awareness of breaking social norms publicly was inhibiting the conduct of snacks.

"It's going to be an inactive morning because I've got some zooms calls"

(TA-2004)

Despite these barriers, there was evidence of persistence, with one participant in a virtual meeting turning their camera off so they could conduct an activity snack and another who adopted an attitude of acceptance towards being seen as comical when conducting snacks in front of others.

"I was just sort of walking up and down the hallway and when I got to the end of the hallway I was doing like a little turning jump. Everyone thought it was kind of amusing but hey."

(TA-2023)

Activities such as sitting in the car, being in bed, crafting, cooking, eating, and conversing with friends, were identified as limiting Snacktivity™, as can be seen with the example of sewing, this could be quite specific to the activity.

"So I don't mind jumping up every now and then and just running up and down the stairs whereas with sewing if I have to jump up I lose my needle and I lose my thread and it's a little bit different then."

(TA-2014)

When activity snacks were difficult to perform due to environmental constraints they were often conducted later in the day. Familial and social engagements such as looking after children and spending time with family also inhibited completing activity snacks.

"Sorry for the silence today but it's a Wednesday and we have our two young grandchildren on that day so I have been ignoring any prompts"

(TA-2021)

The time of day presented a barrier to activity, with several participants identifying that they felt disinclined to do activity snacks early in the day or in the evening.

"I would think from about ten o'clock through till six I will do my snacks, it's unlikely that I'll do many of them after that."

(Int1-2004)

Outdoor activity snacks were reported to be difficult to do due to poor weather and the COVID-19 national lockdown restrictions.

## Routine and habit formation for Snacktivity™

There were indicators that habits were being established and it was apparent that a degree of behavioural persistence had developed.

"A positive point is that I didn't actually think about it this morning, not coming for the walk wasn't even an option, as down as I still feel and as low as I still feel. I didn't

really think I can't be bothered. It has almost become my routine which is really positive."

(TA-2023)

"I think some of them become automatic, definitely"

(Int2-2004)

**Experiences with Snacktivity technology.** Many participants discussed the Fitbit/Snacktivity™ clockface and SnackApp, including the interaction between the technology.

"oh your device hasn't been synched for twenty four hours! I have had a good look but to be honest with you, it seems better to go on to your phone and actually look on the Fitbit app and the Snacktivity app rather than the actual watch"

(Int1-2012)

## Fitbit activity monitor and SnackApp

The Fitbit was popular and considered to be straightforward to use.

". . .getting to grips with the app and the Fitbit which is different to the fitness tracker I normally use—I normally use an Apple watch. I'm finding it good actually. . ."

(TA-2004)

Several participants indicated that the SnackApp features were engaging, such as the step counter, which a couple of participants noted recorded more steps than their phone-based pedometer. Participants' commentary regarding the SnackApp focused primarily on challenges and possible improvements to the application (see below).

"But I'm not having an unmitigated success with the App."

(Int1-2004)

Some participants found that they had difficulty setting up the SnackApp or changing the data displayed on the app, whereas others expressed concerns regarding their understanding of how the SnackApp worked, including how long an activity snack needed to be in duration for it to be recognised by the SnackApp. By far the most problematic element was the recognition of activity snacks by the SnackApp. One participant found the set up and integration of the technology stressful.

"well, I'm annoyed because it doesn't record the activities that I am doing. So for today I've done 3 and its only showing 1. Which is annoying when you feel as though you are flat on your back after you've been running up and down the stairs 50 times."

(TA-2014)

"I've avoided it because of the trauma of setting it up."

(Int1-2010)

Some limits to the ability of the SnackApp to alter behaviour were apparent with one participant choosing to turn it off and another used the calorie counting function as a justification to eat more food. As a result of the challenges experienced with the SnackApp, one participant expressed concerns that future participants may not get sufficient training in how to use it. Further suggestions for improving the SnackApp included broadening its remit to include diet and weight loss and adding a function to record the times that activity snacks were done. One participant suggested adding a button to notify the SnackApp when an activity snack was being conducted, and another suggested including a function to notify the SnackApp of the type of physical activity performed.

## Notification, alerts and prompts

Participants expressed they found that the SnackApp and Fitbit alerts prompted them to engage with Snacktivity™, although they sometimes arrived at inopportune times.

"... I think the reminders at the time are what works for me, rather than setting goals."

(Int1-2021)

"My Fitbit annoys me because it keeps buzzing when I'm in the middle of something, I've started talking to it and telling it to shut up and leave me alone."

(TA-S2014)

When unable to respond to a specific notification some participants chose to defer it,

"I try to respond to the request to do an activity snack, I try to respond there and then. If I can't, if I'm doing something, then I . . . as soon as I've finished doing whatever it is, peeling the potatoes or whatever, I then go outside and do the walking snack."

(Int1-2008)

The relationship with the SnackApp notifications was complex, with participants expressing feelings of annoyance and/or guilt when prompts were not acted upon, one even reporting that they felt that notifications became more insistent the more they ignored them. Suggested improvements included programmable notifications to prompt preferred physical activity at convenient times, with prompts that encourage variety within activity snacks.

## Discussion

Guidelines provide information on the amount of physical activity required for optimal health, but they are not designed to provide the public with the means or motivation to adhere to them. Using a combination of qualitative methods this study found that participants were able to incorporate Snacktivity™ into their daily routines and that they found this approach to promoting physical activity acceptable. Participants commented on the perceived health benefits that they experienced from Snacktivity™, although several barriers to their participation were raised. The technology developed to support and help participants self-monitor their Snacktivity™ was found to be useful, although this was not universal, with some participants experiencing difficulties with set up. Refinements to the Snacktivity™ technology were suggested and participants offered several ways in which this might be achieved.

There was a general sense that participants enjoyed completing Snacktivity™ which is promising, as the way individuals feel about physical activity is an important predictor of whether

they continue to participate [25]. It was encouraging to see that participants reported taking part in over 30 different activity snacks, including both aerobic and strength-based activities while walking-based activity snacks were popular (Table 4). Consistent with self regulation theory and the habit formation model participants reported that they incorporated Snacktivity™ into their existing everyday activities, such as walking for transport, housework and strength exercises [14–16]. Given lack of time is often cited as a barrier to physical activity, adding Snacktivity™ to everyday tasks in this way may make the Snacktivity™ approach time efficient and appealing to the public [7].

The convenience and accessibility of Snacktivity™ was a key factor influencing the location where participants completed their activity snacks. Moreover, in line with self regulation behaviour Snacktivity™ helped participants see how physical activity can be self-directed and convenient. The home was a popular location for Snacktivity™ because it offered a varied environment (e.g., stairs for stepping, walking, walls for press-ups and the kitchen for squats). It was interesting to see that when given the opportunity, participants were creative with developing their ideas for how they were going to complete and integrate Snacktivity™ into their daily lives, using whatever resources were available to them in their environment (e.g. using soft furnishing to perform mountain climbers & using household objects to add resistance to arm training).

Consistent with the strong evidence that has demonstrated the mental and physical benefits of engaging in physical activity participants discussed health benefits that they perceived they had experienced from Snacktivity™, [26, 27]. Our findings that health benefits can be perceived from short bouts of physical activity aligns with evidence supporting the health benefits of the small change approach to promoting health behaviour change [28] and evidence reporting an inverse dose-response relationship between physical activity and all-cause mortality, with any amount of physical activity proving beneficial [29, 30]. Guidance for physical activity states that adults should participate in strength-based physical activity at least twice per week [3]. Of particular concern is that surveillance studies have reported very low levels of participation in strength-based physical activity, indicating that this behaviour may be more difficult to change [31]. Participants reported that they were able to incorporate strength-based Snacktivity™ into their lives, which may prove to be an important benefit of the approach. This is important because strength-based physical activity is known to decrease the risk of falls and osteoporosis [32].

The idea that short bouts of physical activity can have health benefits is not new, but thus far it is not a message that has been promoted to the public because of insufficient high-quality evidence that it leads to long-term changes in physical activity or health. Snacktivity™ has the potential to provide health benefits to the public and future research should now seek to quantify these health benefits.

An important aim of this study was to understand the challenges that participants experienced in performing Snacktivity™ regularly, to help develop and refine this approach. Whilst participants reported engaging in many types of activity snacks, they also reported 'running out of ideas'. Findings highlight that it may be difficult for people to achieve MVPA in 'snack sizes' over longer periods of time and future research should consider how these barriers can be reduced or mitigated. Participants were recruited for this study during the COVID-19 lockdown in England and as such there was a strong focus on completing Snacktivity™ at home which may have limited the nature of the activity snacks that could be achieved. Participants commented that it was more difficult to complete Snacktivity™ at home in the company of others and for the Snacktivity™ approach to reach its potential there may need to be a shift in social norms whereby it viewed as a usual part of everyday life for the public.

Snacktivity™ is based on the principles of the habit formation model and the notion that over time people develop the habit of completing activity snacks each day, which become

established within their daily routines [15, 33]. Small and simple actions may have the potential to become embedded within usual daily routines more quickly than longer or more complex ones. Consistent with the theoretical basis of Snacktivity™, participants referred to a process 'getting in the habit of doing their Snacktivity™' and that this was one of the strategies they used to try and begin to integrate it into their lives, highlighting the important role that habit formation may have for the successful implementation of Snacktivity™.

Technology has been reported to be useful in facilitating health behaviour change and a range of views were expressed about the technology offered to support Snacktivity™ [34]. Participants reported that the push notification successfully prompted them to engage in Snacktivity™, which is consistent with other studies that have reported nudges to be useful in facilitating behaviour change [35]. Whilst the views expressed about the Snacktivity™ technology were generally positive, some frustrations with the process of setting up the SnackApp and Fitbit were reported. Findings highlighted that whilst technology is an integral part of life, many people continue to experience difficulties in using technology meaning that they are unable to benefit from using it, potentially undermining their motivation. Of note, when participants were asked to suggest ideas of how the Snacktivity™ intervention could be improved, their feedback was predominately centred around improvements to the technology, rather than other components of the intervention.

This study had several strengths and limitations that should be considered when interpreting the findings. To our knowledge, this is the first study to investigate participants experiences of participating in Snacktivity™ over time, whilst using a combination of qualitative research methods to offer an in-depth understanding of experiences. Rather than collecting data from a single interview at one point in time, we interviewed participants on two occasions and included a think-aloud component in the research to add further richness to the data (included 103 recordings). Participants from a range of different socio-economic backgrounds were recruited. A comprehensive approach to analysis was used, triangulating data collected from two qualitative methods. Three researchers with different expertise and occupational backgrounds were involved in the analysis ensuring the results were guided by a range of perspectives and experiences.

## Potential limitations of this study

The study was conducted during the COVID-19 lockdown period in England and participants mostly completed their Snacktivity™ at home therefore this study may underreport the wider possibilities of the Snacktivity™ approach. The work environment may also provide many opportunities for Snacktivity™. About 40% of participants were retired and thus findings may align less well with younger age groups. We did not aim to reach data saturation per se meaning some information relevant to this study may not have been harnessed. Rather, the goal was to interview all the participants who consented to be interviewed to achieve information power [36].

## Conclusion

The public may be open to exploring opportunities for Snacktivity™ within their lives so that they can experience the health benefits that participation in regular short bouts of physical activity throughout the day may have to offer. Following the extensive development work conducted here and elsewhere the Snacktivity intervention is ready for evaluation in a full-scale randomised controlled trial.

## Supporting information

**S1 Checklist. Consolidated criteria for reporting qualitative research checklist.**
(PDF)

**S1 File. Study interview schedule.**
(DOCX)

**S2 File. Theme coding process.**
(DOCX)

## Acknowledgments

The authors would like to thank the Public Advisory Group (PAG) who contributed to this research. The Snacktivity™ Public Advisor Group (PAG) were involved in the development of this study, particularly in shaping the participant facing information, which included guidance for conducting the think aloud study. The authors would like to thank Birmingham Community Healthcare Foundation NHS Trust, the Clinical Research Network (West Midlands) and the general practices who supported the study. The authors would like to thank all those who participated in this research.

## Author Contributions

**Conceptualization:** Sheila M. Greenfield, Amanda J. Daley.

**Data curation:** Matthew Krouwel, Anna Chalkley.

**Formal analysis:** Matthew Krouwel, Tom Yates.

**Funding acquisition:** Sheila M. Greenfield, Anna Chalkley, Kate Jolly, Magdalena Skrybant, Stuart J. H. Biddle, Colin Greaves, Ralph Maddison, Nanette Mutrie, Natalie Ives, Dale W. Esliger, Lauren Sherar, Charlotte L. Edwardson, Emma Frew, Sarah Tearne, Amanda J. Daley.

**Investigation:** Anna Chalkley, Amanda J. Daley.

**Methodology:** Sheila M. Greenfield, Helen M. Parretti, Kate Jolly, Magdalena Skrybant, Stuart J. H. Biddle, Colin Greaves, Ralph Maddison, Nanette Mutrie, Amanda J. Daley.

**Project administration:** Matthew Krouwel, Anna Chalkley, Amanda J. Daley.

**Resources:** Amanda J. Daley.

**Software:** Matthew Krouwel, Anna Chalkley, Amanda J. Daley.

**Supervision:** Amanda J. Daley.

**Validation:** Matthew Krouwel, Sheila M. Greenfield, Amanda J. Daley.

**Visualization:** Matthew Krouwel, Amanda J. Daley.

**Writing – original draft:** Matthew Krouwel, Amanda J. Daley.

**Writing – review & editing:** Anna Chalkley, James P. Sanders, Helen M. Parretti, Kajal Gokal, Kate Jolly, Magdalena Skrybant, Stuart J. H. Biddle, Colin Greaves, Ralph Maddison, Nanette Mutrie, Natalie Ives, Dale W. Esliger, Lauren Sherar, Charlotte L. Edwardson, Tom Yates, Emma Frew, Sarah Tearne.

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
