## [Decision Letter · Decision Letter 0]

19 Jun 2023

PONE-D-23-08763Promoting participation in physical activity through Snacktivity™: A qualitative mixed methods studyPLOS ONE

Dear Dr. Daley,

Thank you for submitting your manuscript to PLOS ONE. After careful consideration, we feel that it has merit but does not fully meet PLOS ONE’s publication criteria as it currently stands. Therefore, we invite you to submit a revised version of the manuscript that addresses the points raised during the review process

The manuscript is interesting and needs several corrections to be implemented to make it suitable for publishing.

Abstract

Line 56-59 this should be part of the methodology and not the background of the abstract.

“We have proposed a whole-day approach to promoting physical activity called Snacktivity™, which encourages frequent 2-5 minute ‘activity snacks’ of moderate-to-vigorous intensity. Using

repeated semi-structured interviews and a think aloud protocol,”

Methodology

Line 130-131

“Participants were recruited to take part in a study that involved both qualitative and quantitative data collection. The report focuses on the qualitative data only. As part of the quantitative data collection procedures, participants completed assessments of health outcomes at the start and end of a three-week Snacktivity™ intervention period (data not reported here).”

Why was the quantitative data not reported alongside the qualitative study? This could have expressed the quality of the study giving more explanation based on the findings which will be more appreciated with regards to the same topic. I suggest you include the quantitative aspect of the study which will be better or entirely remove the methodological aspect of the quantitative study in this report.

Full what is meaning of AD in the study? Please clearly state before using it in the manuscript

Results

Recruitment and data

Line 233-236 This part of the report should be under materials and methods

“A total of 583 invitations were sent to patients using an NHS podiatry service and 47 sent

to general practice patients who had participated in an earlier Snacktivity™ study. From these

invitations, 20 eligible individuals agreed to take part (6 men & 14 women) in the larger study. No data was gathered regarding reasons for non-participation”

The author can start the sentence with “eleven out of the 20 participants…….”

Line 261 (Table 2)

Table 2 – Themes and subthemes across all three analysis and combined themes

For clarity of the Table, I request that each activity (the 3 interviews and combined data table) has a Table, the themes generated, the subthemes, generated codes and repetition of code by each participant for elaboration on the data analysis conducted.

The translation is not clear, the authors may need to do some punctuations and the grammar to ensure that the person reading it understands. Eg line 270-274. This should be done where required in the whole manuscript.

 “You know people always think, you know, you always think, oh you know you need to go and do a twenty minute workout, and I think you know what you’ve shown us is that we can you know choose to do whatever, whatever it is that you enjoy doing, just … just have that moment for two minutes, you know, it doesn’t even have to be five ...”(int2-2006)

Discussion

The discussion should not have headings rather, you can use a paragraph for each point that was raised in the discussion. Furthermore, you can discuss the strength of the study under a paragraph while the limitations of the study should be the last paragraph and stand alone.

We look forward to receiving your revised manuscript.

Kind regards,

Ayi Vandi Kwaghe, D.V.M., M.V.Sc., P.G.D.E. Ph.D., MPH

Academic Editor

PLOS ONE

Journal Requirements:

The Snacktivity™ trial is funded by the NIHR (Programme Grants for Applied Research, reference number: RP-PG-0618–20008). AJD is supported by a National Institute for Health Research (NIHR) Research Professorship award. KJ and SG are part-funded by NIHR Applied Research Collaboration (ARC) West Midlands. This research was supported by the NIHR Leicester Biomedical Research Centre. This publication presents independent research funded by the NIHR. The views expressed are those of the author(s) and not necessarily those of the NHS, the NIHR or the Department of Health and Social Care.

However, funding information should not appear in the Acknowledgments section or other areas of your manuscript. We will only publish funding information present in the Funding Statement section of the online submission form. 

This work was supported by the National Institute for Health Research (NIHR,

www.nihr.ac.uk.) (grant reference RP-PG-0618-20008). AJD is supported by a National

Institute for Health Research (NIHR, www.nihr.ac.uk.) Research Professorship award.

The funders had no role in study design, data collection and analysis, decision to

publish, or preparation of the manuscript

4. Please amend the manuscript submission data (via Edit Submission) to include author Sheila M Greenfield.

Reviewers' comments:

Reviewer's Responses to Questions

**Comments to the Author**

1. Is the manuscript technically sound, and do the data support the conclusions?

Reviewer #1: Yes

Reviewer #2: Partly

2. Has the statistical analysis been performed appropriately and rigorously? 

Reviewer #1: Yes

Reviewer #2: N/A

3. Have the authors made all data underlying the findings in their manuscript fully available?

Reviewer #1: No

Reviewer #2: No

4. Is the manuscript presented in an intelligible fashion and written in standard English?

Reviewer #1: Yes

Reviewer #2: Yes

5. Review Comments to the Author

Reviewer #1: This was a very well-written and interesting paper, and this type of rich user experience data for a possible intervention is extremely important and valuable. I have a few minor comments.

I would have liked to see the interview schedule(s) included as supplementary material

A little more context in the methods about the two sets of qualitative interviews would be useful (e.g. the process and the rationale for considering these are separate and the decision around timing of these). My initial reading from the methods made me think they were repeat interviews with the same participants, but the results suggested they were different participants. Clarity around this would be helpful.

I wondered if it the table could be reformatted slightly. I realise that is difficult when presenting multiple results, but I didn't not find it very easy to read in its current format. Perhaps (e.g.) a standalone table with the condensed themes might be useful.

Reviewer #2: Dear colleagues,

-The Snacktivity approach seems remarkable and like it has great promise! The think aloud component in particular is a strength of this study. I’d like to encourage the authors to cite some more recent literature, and to go a little more in-depth in some areas. As a whole though, this is a strong paper. I have included comments below that should help strengthen this paper.

Abstract:

How were eleven adults interviewed twice to equal 21?

I’d argue that they weren’t “prevented from” exercising in front of others but chose not to.

Acknowledging the timeframe of the study would be useful; pre-COVID-19, doing activity at home was not as accessible as during the timeframe of 2020-December 2021 when working from home was more common. This should also be recognized in presenting the results (e.g., relevant context).

Literature Review:

Line 104: while it may be a (perceived) lack of time for some, it is truly a lack of time for others. Perhaps “often” could be added to the parentheses to avoid assigning blame.

Line 105: many of the citations are dated 2007/2010; anything more recent? There have been many more increased demands on people’s times over the past 15 years, particularly given the numerous recessions and economical disasters (plus things like COVID-19).

Methods:

The title leads me to a bit of confusion – how is it both qualitative and mixed-methods? Wouldn’t it just be mixed-methods? But, since only the qualitative portions are described here, it should be simply qualitative right?

Why did respondents have to be “inactive or moderately inactive” vs already active? It seems that the goal was to increase physical activity, specifically using Snacktivity, so could someone who was already active continue to add some habit-forming behaviors and Snacktivity use in too?

I’m curious as to why podiatry was the specialty that was focused on. Can the authors explain this further?

The interviews are a bit on the shorter side. Do the authors still feel like they got everything they could out of the participants related to the topics of interest?

What was the timeline of the total intervention to completion of the second interview? Could there be a recall bias at all (it doesn’t sound like it from the manuscript as written); just checking.

Results:

It’s interesting that the second interview study was where “awareness and impact of being a participant in a health study” really emerged for the first time. And why was it combined into the larger set of three themes? It seems pretty distinctive to me. Combining the others makes sense; this is the one I think deserves some attention on its own.

Why is “stair climbing” capitalized in Table 3?

Going back to the “feeling like they couldn’t snack publicly” – what types of employment did the participants have? I know many people who take video calls while on the treadmill or bike at home or in the office so this is a surprising finding.

It’s interesting that children are an excuse to NOT complete the snacks. Children, especially small ones, should likely be the ones to encourage engagement in jumping and moving! I have young children myself and I find that I am the only one who limits myself with working out around them for short bursts; they want to participate! There may not be anything to add here for the paper but just a thought.

What does the participant mean, the “trauma” of setting up an app? How is setting up an app traumatic? Now, perhaps this is due to the age of the participant? I’m interested in knowing more.

Discussion:

It is mentioned early on the self-regulation theory and the habit formation model provided the basis for the intervention. Yet, it is not discussed in relationship with the interview or think-aloud findings, nor the implications related. Please expand.

Lines 453-454: In what ways were the findings not universal? Any common factors between the individuals?

6. PLOS authors have the option to publish the peer review history of their article (what does this mean?). If published, this will include your full peer review and any attached files.

Reviewer #1: No

Reviewer #2: **Yes: **Grace Ellen Brannon

---

## [Author Response · Author response to Decision Letter 0]

24 Jul 2023

Note: Line numbers refer to the tracked changed version of the manuscript for ease of reading

Editor requested edits

Abstract

Line 56-59 this should be part of the methodology and not the background of the abstract.

RESPONSE: This has been amended.

Line 130-131: “Participants were recruited to take part in a study that involved both qualitative and quantitative data collection. The report focuses on the qualitative data only. As part of the quantitative data collection procedures, participants completed assessments of health outcomes at the start and end of a three-week Snacktivity™ intervention period (data not reported here).” Why was the quantitative data not reported alongside the qualitative study? This could have expressed the quality of the study giving more explanation based on the findings which will be more appreciated with regards to the same topic. I suggest you include the quantitative aspect of the study which will be better or entirely remove the methodological aspect of the quantitative study in this report.

RESPONSE: Reference to the quantitative component of the study has been removed.

Full what is meaning of AD in the study? Please clearly state before using it in the manuscript.

RESPONSE: AD refers to the author Amanda Daley.

Recruitment and data: Line 233-236 This part of the report should be under materials and methods

RESPONSE: This has been changed.

“A total of 583 invitations were sent to patients using an NHS podiatry service and 47 sent to general practice patients who had participated in an earlier Snacktivity™ study. From these invitations, 20 eligible individuals agreed to take part (6 men & 14 women) in the larger study. No data was gathered regarding reasons for non-participation”. The author can start the sentence with “eleven out of the 20 participants…….”

RESPONSE: This has been corrected (Lines 234-240).

Line 261 (Table 2). Table 2 – Themes and subthemes across all three analysis and combined themes. For clarity of the Table, I request that each activity (the 3 interviews and combined data table) has a Table, the themes generated, the subthemes, generated codes and repetition of code by each participant for elaboration on the data analysis conducted.

RESPONSE: To prevent an excess of tables in the main text, the tables of themes generated for each of the studies/datasets has been included and presented in a supplementary file.

The translation is not clear, the authors may need to do some punctuations and the grammar to ensure that the person reading it understands. Eg line 270-274. This should be done where required in the whole manuscript. “You know people always think, you know, you always think, oh you know you need to go and do a twenty minute workout, and I think you know what you’ve shown us is that we can you know choose to do whatever, whatever it is that you enjoy doing, just … just have that moment for two minutes, you know, it doesn’t even have to be five ...”(int2-2006)

RESPONSE: We have made some minor changes to punctuations and grammar where we can but we wanted to highlight that we are reporting direct speech so it is not always possible to change the text without misreporting what has been said by participants. As is usual practice, occasionally with have used the text (…….) to denote where irrelevant text has been omitted. 

The discussion should not have headings rather, you can use a paragraph for each point that was raised in the discussion. Furthermore, you can discuss the strength of the study under a paragraph while the limitations of the study should be the last paragraph and stand alone. Please ensure that your decision is justified on PLOS ONE’s publication criteria and not, for example, on novelty or perceived impact.

RESPONSE: The discussion has been edited as requested.

Reviewer #1: This was a very well-written and interesting paper, and this type of rich user experience data for a possible intervention is extremely important and valuable. I have a few minor comments.

RESPONSE: We would like to thank the reviewer for their kind words.

I would have liked to see the interview schedule(s) included as supplementary material

RESPONSE: The interview schedule has been added as supplementary file.

A little more context in the methods about the two sets of qualitative interviews would be useful (e.g. the process and the rationale for considering these are separate and the decision around timing of these). My initial reading from the methods made me think they were repeat interviews with the same participants, but the results suggested they were different participants. Clarity around this would be helpful.

RESPONSE: The participants in interviews 1 & 2 were the same people to capture change over time, with the exception of S2008, who was unable to conduct the second interview. We have clarified this on line 194 and in the abstract. 

I wondered if it the table could be reformatted slightly. I realise that is difficult when presenting multiple results, but I didn't not find it very easy to read in its current format. Perhaps (e.g.) a standalone table with the condensed themes might be useful.

RESPONSE: We have amendment the theme table (Table 2) in the main text and we hope this is clearer now. 

Reviewer #2

The Snacktivity approach seems remarkable and like it has great promise! The think aloud component in particular is a strength of this study. I’d like to encourage the authors to cite some more recent literature, and to go a little more in-depth in some areas. As a whole though, this is a strong paper. I have included comments below that should help strengthen this paper.

RESPONSE: We would like to thank the reviewer for their kind words and feedback.

Abstract:

How were eleven adults interviewed twice to equal 21?

RESPONSE: One participants did not complete a second interview. This has been clarified in the abstract and main text (Line 71 and Line 194). 

I’d argue that they weren’t “prevented from” exercising in front of others but chose not to.

RESPONSE: We appreciate this feedback. We have tried to present findings as expressed by participants and have added the word “perceived” to help clarify this statement

Acknowledging the timeframe of the study would be useful; pre-COVID-19, doing activity at home was not as accessible as during the timeframe of 2020-December 2021 when working from home was more common. This should also be recognized in presenting the results (e.g., relevant context).

RESPONSE: Due to the abstract word limit we are not able to add any further information. However, the methods and discussion acknowledge that the study too place during COVID-19.

Literature Review:

Line 104: while it may be a (perceived) lack of time for some, it is truly a lack of time for others. Perhaps “often” could be added to the parentheses to avoid assigning blame.

RESPONSE: This line has been edited as suggested by the reviewer-thank you (Line 105).

Line 105: many of the citations are dated 2007/2010; anything more recent? There have been many more increased demands on people’s times over the past 15 years, particularly given the numerous recessions and economical disasters (plus things like COVID-19).

RESPONSE: Some references have not been updated as they are seminal papers. However, two references have been updated. (references 7 & 9). 

Methods:

The title leads me to a bit of confusion – how is it both qualitative and mixed-methods? Wouldn’t it just be mixed-methods? But, since only the qualitative portions described here, it should be qualitative right?

RESPONSE: Yes we believe the title is correct as we have mixed methods by way of the inclusion of the interview study and think aloud study.

Why did respondents have to be “inactive or moderately inactive” vs already active? It seems that the goal was to increase physical activity, specifically using Snacktivity, so could someone who was already active continue to add some habit-forming behaviors and Snacktivity use in too?

RESPONSE: We agree that Snacktivity™ may be useful for all citizens regardless of their physical activity but we believe those who are inactive are likely to have the most to gain. For this reason we focused on citizens who were the most inactive. 

I’m curious why podiatry was the specialty that was focused on. Can the authors explain this further?

RESPONSE: A community health trust is supporting the trial which has a large and research ready podiatry service and therefore we decided to use this service for this study.

The interviews are a bit on the shorter side. Do the authors still feel like they got everything they could out of the participants related to the topics of interest?

RESPONSE: A policy of allowing the interviews to be succinct was followed to avoid overburdening participants and thus reducing the likelihood of them declining to participate in the second interview (Lines 198-200). Some participants were also involved in the think aloud study. Whilst the interview were succinct they nevertheless still generated a substantial amount of data for this study, coupled with the think aloud data. 

What was the timeline of the total intervention to completion of the second interview? Could there be a recall bias at all (it doesn’t sound like it from the manuscript as written); just checking.

RESPONSE: Interviews were conducted at the start and end of the three week Snacktivity™ intervention period (Lines193-195).

Results: It’s interesting that the second interview study was where “awareness and impact of being a participant in a health study” really emerged for the first time. And why was it combined into the larger set of three themes? It seems pretty distinctive to me. Combining the others makes sense; this is the one I think deserves some attention on its own.

RESPONSE: Thank you for this feedback. We have commented on the fourth theme in the manuscript (please see lines 250-259). In summary here, the theme ‘awareness and impact of being a participant in a health study’, was not included in the combined themes as it was considered to be primarily of interest to the research team in the methodological design of future Snacktivity™ trials. Information regarding this fourth theme can be found in a supplementary file named ‘Theme coding process’ in Appendix C. 

Why is “stair climbing” capitalized in Table 3?

RESPONSE: Thank you for highlighting this which has been corrected. 

Going back to the “feeling like they couldn’t snack publicly” – what types of employment did the participants have? I know many people who take video calls while on the treadmill or bike at home or in the office so this is a surprising finding.

RESPONSE: We do not know the exact professions of participants only what their status was as described in Table 1a.

It’s interesting that children are an excuse to NOT complete the snacks. Children, especially small ones, should likely be the ones to encourage engagement in jumping and moving! I have young children myself and I find that I am the only one who limits myself with working out around them for short bursts; they want to participate! There may not be anything to add here for the paper but just a thought.

RESPONSE: Thank you for this comment.

What does the participant mean, the “trauma” of setting up an app? How is setting up an app traumatic? Now, perhaps this is due to the age of the participant? I’m interested in knowing more.

RESPONSE: Some of the participants were older adults and as the reviewer highlights some initially had difficulty setting this up. We discuss this point on Lines 522-524.

Discussion: It is mentioned early on the self-regulation theory and the habit formation model provided the basis for the intervention. Yet, it is not discussed in relationship with the interview or think-aloud findings, nor the implications related. Please expand.

RESPONSE: We have a paragraph regarding habit formation in the discussion (Lines 510-517). We had discussed the results in the context of self regulation theory but we have now tried to make this more explicit within the discussion.

Lines 453-454: In what ways were the findings not universal? Any common factors between the individuals?

RESPONSE: We have added some text to this comment to make it clearer (Lines 454-456.)

Journal formatting corrections

Thank you for stating the following in the Acknowledgments Section of your manuscript: The Snacktivity™ trial is funded by the NIHR (Programme Grants for Applied Research, reference number: RP-PG-0618–20008). AJD is supported by a National Institute for Health Research (NIHR) Research Professorship award. KJ and SG are part-funded by NIHR Applied Research Collaboration (ARC) West Midlands. This research was supported by the NIHR Leicester Biomedical Research Centre. This publication presents independent research funded by the NIHR. The views expressed are those of the author(s) and not necessarily those of the NHS, the NIHR or the Department of Health and Social Care. However, funding information should not appear in the Acknowledgments section or other areas of your manuscript. We will only publish funding information present in the Funding Statement section of the online submission form. Please remove any funding-related text from the manuscript and let us know how you would like to update your Funding Statement. Currently, your Funding Statement reads as follows: 

This work was supported by the National Institute for Health Research (NIHR,www.nihr.ac.uk.) (grant reference RP-PG-0618-20008). AJD is supported by a National Institute for Health Research (NIHR, www.nihr.ac.uk.) Research Professorship award. The funders had no role in study design, data collection and analysis, decision to publish, or preparation of the manuscript. Please include your amended statements within your cover letter; we will change the online submission form on your behalf.

RESPONSE: We have included an amended statement of funding in the cover letter.

In your Data Availability statement, you have not specified where the minimal data set underlying the results described in your manuscript can be found. PLOS defines a study's minimal data set as the underlying data used to reach the conclusions drawn in the manuscript and any additional data required to replicate the reported study findings in their entirety. All PLOS journals require that the minimal data set be made fully available. "Upon re-submitting your revised manuscript, please upload your study’s minimal underlying data set as either Supporting Information files or to a stable, public repository and include the relevant URLs, DOIs, or accession numbers within your revised cover letter. We will update your Data Availability statement to reflect the information you provide in your cover letter.

RESPONSE: We have now added the data to the Loughborough University data depository 'Snacktivity' to promote physical activity and reduce future risk of disease in the population (figshare.com). We have also included the data availability statement in the cover letter as requested.

Please amend the manuscript submission data (via Edit Submission) to include Sheila M Greenfield.

RESPONSE: This has been corrected.

Please include captions for your Supporting Information files at the end of your manuscript, and update any in-text citations to match accordingly. Please see our Supporting Information guidelines for more information: http://journals.plos.org/plosone/s/supporting-information.

RESPONSE: Captions for supporting information files have been added at the end of the paper.

RESPONSE: N/A

---

## [Decision Letter · Decision Letter 1]

31 Jul 2023

PONE-D-23-08763R1Promoting participation in physical activity through Snacktivity™: A qualitative mixed methods studyPLOS ONE

Dear Dr. Daley,

Thank you for submitting your manuscript to PLOS ONE. After careful consideration, we feel that it has merit but does not fully meet PLOS ONE’s publication criteria as it currently stands. Therefore, we invite you to submit a revised version of the manuscript that addresses the points raised during the review process.

ACADEMIC EDITOR:The authors have responded well to the questions but still need to do so adjustments to the manuscript as stated below:

Line 261 (Table 2). Table 2 – Themes and subthemes across all three analysis and combined themes. For clarity of the Table, I request that each activity (the 3 interviews and combined data table) has a Table, the themes generated, the subthemes, generated codes and repetition of code by each participant for elaboration on the data analysis conducted.

The response by the authors with regard to the above question is not satisfactory. The authors need to include the codes for each subtheme generated if the codes were used in generating the subthemes before the formation of the themes. That will not in any way congest the table but will make the table more explanatory with regards to the study. Codes are the basis for a qualitative study. Definitions of terms for the table can be done as foot notes of the table.

Kindly state the inclusion and exclusion criteria of the study. The authors stated the eligibility for participation which I assume was the inclusion criteria but what about the exclusion criteria of the study?

Did you reach ‘data saturation point’ while conducting the interviews? If yes, kindly state it in your methods and if you have not reached “data saturation point” clearly state that as part of the limitation of the study in your discussion. This is relevant in any qualitative study, it hints on the fact whether enough information has been gathered during the study or there is more information that could be explored in the area of the study which has not been harnessed

We look forward to receiving your revised manuscript.

Kind regards,

Ayi Vandi Kwaghe, D.V.M., M.V.Sc., P.G.D.E. Ph.D., MPH

Academic Editor

PLOS ONE

Journal Requirements:

Reviewers' comments:

Reviewer's Responses to Questions

**Comments to the Author**

1. If the authors have adequately addressed your comments raised in a previous round of review and you feel that this manuscript is now acceptable for publication, you may indicate that here to bypass the “Comments to the Author” section, enter your conflict of interest statement in the “Confidential to Editor” section, and submit your "Accept" recommendation.

Reviewer #1: All comments have been addressed

Reviewer #2: (No Response)

2. Is the manuscript technically sound, and do the data support the conclusions?

Reviewer #1: (No Response)

Reviewer #2: Yes

3. Has the statistical analysis been performed appropriately and rigorously? 

Reviewer #1: (No Response)

Reviewer #2: N/A

4. Have the authors made all data underlying the findings in their manuscript fully available?

Reviewer #1: (No Response)

Reviewer #2: Yes

5. Is the manuscript presented in an intelligible fashion and written in standard English?

Reviewer #1: (No Response)

Reviewer #2: Yes

6. Review Comments to the Author

Reviewer #1: (No Response)

Reviewer #2: I appreciate the authors' efforts in responding to my own and other reviewers' recommendations and comments. A few follow-ups:

1) The title should really be "Promoting participation in physical activity through Snacktivity™: A mixed methods study" based on the explanation provided. As in, the authors describe the inclusion of the interview study (qualitative) and the think aloud study (voice notes; really this is qualitative) (and I'll add, each of those based on the intervention components). Therefore, the title should not be "qualitative mixed methods" but simply "mixed methods" and mixed methods inherently encompasses both qual and quant research.

2) Since the exact professions of participants were not collected, this could be discussed as a limitation or future direction for this stream of research.

3) Re: the "trauma" of setting up the app; I don't actually see the point discussed on lines 522-524 (the original line is 418). Can the authors clarify if this change was made in this version of the submission? I tried searching through the document using keywords and Ctrl-F but it was not successful.

4) Similarly, re: self-regulation theory and habit formation, lines 510-517 are referenced as where the discussion should describe more explicitly. It looks like those are not the correct line numbers (I see lines 484-485 and 491-492 now name the theory and model); it also seems like the authors simply state that the present study's findings are "consistent with" or "in line with"...can the authors describe HOW this study is in line with or consistent with? Does this study expand previous research at all? Or simply confirm? If it doesn't expand, in what ways in this study adding to existing knowledge? Please outline the specific contributions.

7. PLOS authors have the option to publish the peer review history of their article (what does this mean?). If published, this will include your full peer review and any attached files.

Reviewer #1: **Yes: **Abi Fisher

Reviewer #2: No

---

## [Author Response · Author response to Decision Letter 1]

17 Aug 2023

The authors have responded well to the questions but still need to do so adjustments to the manuscript as stated below:

RESPONSE: We are pleased that we have responded well to the questions raised.

Line 261 (Table 2). Table 2 – Themes and subthemes across all three analysis and combined themes. For clarity of the Table, I request that each activity (the 3 interviews and combined data table) has a Table, the themes generated, the subthemes, generated codes and repetition of code by each participant for elaboration on the data analysis conducted. The response by the authors with regard to the above question is not satisfactory. The authors need to include the codes for each subtheme generated if the codes were used in generating the subthemes before the formation of the themes. That will not in any way congest the table but will make the table more explanatory with regards to the study. Codes are the basis for a qualitative study. Definitions of terms for the table can be done as foot notes of the table.

RESPONSE: We have provided codes, sub themes and main themes in Table 2 (lines 265-266) in the manuscript and directed readers to supplementary file 4 for further information on theme generation according to each data set and the combined data set used in the study (see Lines 254-255). Additionally, within supplementary file 4 we have provided information for all three individual data sets and the combined themes regarding; main themes, sub themes, codes, ID for participants who contributed to every code and the number of references for a code that were recorded. We believe that collectively this information provides a substantial amount of transparency to readers regarding the process by which the main themes were generated. We believe this is the best approach to provide readers with the information required to assess the methods and findings of the manuscript, whilst keeping the information in the manuscript manageable for readers to digest.

Kindly state the inclusion and exclusion criteria of the study. The authors stated the eligibility for participation which I assume was the inclusion criteria but what about the exclusion criteria of the study?

RESPONSE: Both inclusion and exclusion criteria are included now (see Lines 153-156).

Did you reach ‘data saturation point’ while conducting the interviews? If yes, kindly state it in your methods and if you have not reached “data saturation point” clearly state that as part of the limitation of the study in your discussion. This is relevant in any qualitative study, it hints on the fact whether enough information has been gathered during the study or there is more information that could be explored in the area of the study which has not been harnessed

RESPONSE: We addressed the issue of saturation in the previous submission but we have added some additional text to the methods and discussion sections (Lines 156-159 and 546-549).

---

## [Editor Report · Decision Letter 2]

22 Aug 2023

Promoting participation in physical activity through Snacktivity™: A qualitative mixed methods study

PONE-D-23-08763R2

Dear Dr. Daley,

We’re pleased to inform you that your manuscript has been judged scientifically suitable for publication and will be formally accepted for publication once it meets all outstanding technical requirements.

Kind regards,

Ayi Vandi Kwaghe, D.V.M., M.V.Sc., P.G.D.E. Ph.D., MPH

Academic Editor

PLOS ONE

Additional Editor Comments (optional):

The authors have responded well to the corrections they were asked to implement. Qualitative studies are based on collecting in-depth information on a subject matter. However, on the issue of "data saturation point" you should not decide not to put it into consideration as stated in your manuscript. It is what you observe based on the data collected from your interviews. Were there repetitions in the data collected that you feel there is no new information on the subject matter or there is more to be gathered because more information was gathered with out reaching "data saturation point"? It is a matter of statement based on the study conducted. I suggest that you state it as requested. This may serve as a link to other researchers to explore on the subject matter in the future.

"155 this study was modest, it was decided that all participants who consented would be interviewed

156 and data would be analysed for maximum richness, without reference to concepts of saturation."
---

## [Editor Report · Acceptance letter]

1 Sep 2023

PONE-D-23-08763R2 

Promoting participation in physical activity through Snacktivity: A qualitative mixed methods study 

Dear Dr. Daley:

I'm pleased to inform you that your manuscript has been deemed suitable for publication in PLOS ONE. Congratulations! Your manuscript is now with our production department. 

Kind regards, 

on behalf of

Dr. Ayi Vandi Kwaghe 

Academic Editor

PLOS ONE